# A New Numerical Finite Strain Procedure for a Circular Tunnel Excavated in Strain-Softening Rock Masses and Its Engineering Application

Wenbo Chen [ID], Dingli Zhang *, Qian Fang, Xuanhao Chen and Tong Xu

Key Laboratory for Urban Underground Engineering of Ministry of Education, Beijing Jiaotong University, Beijing 100044, China; 19115010@bjtu.edu.cn (W.C.); qfang@bjtu.edu.cn (Q.F.); 19115011@bjtu.edu.cn (X.C.); 16115280@bjtu.edu.cn (T.X.)
* Correspondence: dlzhang@bjtu.edu.cn; Tel.: +86-10-5168-8111

**Abstract:** The small strain theory underestimates the self-bearing capacity of rock masses, especially for a soft rock tunnel under high geostress. To perform an efficient and accurate calculation and provide a reference for the stiffness design of a tunnel, the finite strain solution for a circular tunnel in Mohr–Coulomb strain-softening rock masses with a non-associated flow rule was derived as three sets of differential equations under the Lagrangian coordinate, which are in the residue region, the softening region, and the elastic region, respectively. Based on the bisection method, an iteration procedure for solving the finite strain solution was proposed to approximate the boundary condition at infinity, the values of two adjacent boundaries, and the initial values on the excavation boundary. This numerical procedure was verified by comparing with self-similar solutions, recursive solutions, and FLAC simulation results. In the calculation example, the relative error on boundaries can be decreased to less than $10^{-8}$ after only 10 times iteration and the time for each calculation is less than 15 s. Applying this procedure on the sensibility analysis and stiffness reliability design for the Zhongyi tunnel, a support stiffness of 4.3 MPa/m is recommended to guarantee a tunnel displacement lower than 0.5 m.

**Keywords:** finite strain; circular tunnel; strain-softening rock masses; global sensibility analysis; stiffness design

## 1. Introduction

With the rapid construction of transportation infrastructure, more and more tunnels have been built in soft rock under high geostress. Large deformation occurs frequently in such tunnel engineering and even causes roof collapse, steel ribs' twisting and breaking, shotcrete cracking, lining cracking, the invasion of primary lining into the space of secondary lining, and other disasters [1]. For instance, the maximum deformation of the Muzhailing Tunnel located upon the Lanzhou–Haikou National Expressway exceeds 2000 mm, far exceeding the deformation allowance [2]. Similar problems are encountered in the Muzhailing Railway Tunnel [3], Zhegu Mountain Tunnel [4], Huangjiazhai Tunnel [5], Minxian Tunnel [6] and have also been predicted to appear during the construction of 43 soft rock tunnels with high geostress in the Sichuan–Tibet railway [7]. The large deformation hazard raises significant challenges for the design of the tunnel support system.

In recent years, scholars have been conducting considerable research on the mechanical calculation of tunnels excavated in strain-softening rock masses. Alonso et al. [8] and Park [9] derived a similarity solution for the circle tunnel in strain-softening rock mass, which it still needed to be solved by a numerical method. Guan et al. [10] compared a simplified theoretical method with a rigorous theoretical method for tunnel excavation in the rock masses exhibiting strain-softening behavior. Park et al. [11], Lee et al. [12], Wang et al. [13], Zhang et al. [14], Han et al. [15], Cui et al. [16], Zou et al. [17], Wang et al. [18], Zhang et al. [19], Xia et al. [20],

and GhorbaniHadi et al. [21] proposed a series of new semi-analytical solutions and numerical procedures for calculating the stress and displacement around a circular tunnel excavated in strain-softening Mohr–Coulomb, generalized Hoek–Brown, or three-dimensional nonlinear Hoek–Brown rock mass. Although these methods were applied well on the mechanical analysis and the tunnel design [16,22–27], they are based on the small deformation hypothesis and not applicable to analyze a tunnel with a high risk of large deformation.

For the large strain problem, Durban [28], Yu and Rowe [29], and Vrakas [30,31] derived finite strain closed-form solutions of circular opening issues in the elastic and the elasto-perfectly-plastic continuous medium based on a geometric equation in Lagrangian coordinates [32]. Park [33] proposed a large strain similarity solution, expressed as first-order ordinary differential equations, for a spherical or circular opening in elasto-perfectly-plastic rock masses. Mo et al. [34] described the confinement-convergence responses for deep tunnels in linearly elastic, non-associated MC, and brittle Hoek–Brown rock masses using large-strain solutions of unified spherical and cylindrical cavity contraction. Zhang et al. [35] derived a large strain differential equations for elasto-brittle-plastic rock masses and implemented a numerical procedure.

For a tunnel in strain-softening rock masses, Guan et al. [36] analyzed a highly deformed circular tunnel in the Lagrangian coordinate by dividing the plastic zone into $n$ concentric annuli and using analytical solutions for the elastic-brittle-plastic rock mass within each annulus. Zhang et al. [37] derived the differential equations in the original coordinate by the stress equilibrium equation and deformation compatibility equation and proposed one numerical procedure which updated the variables of $(i + 1)$th annulus in the softening region by the fifth-order Runge–Kutta method according to the data of the $i$th annulus. Then Zhang et al. [38] converted differential equations into integral equations and implemented a recursive procedure named 'softening annulus loop'. In a similar method, Xu et al. [39] developed a numerical solution for strain-softening rock masses obeying the GZZ strength criterion. The recursion method in [36–39] inevitably leads to an accumulative error which cannot be evaluated quantitatively and adjusted automatically. Increasing the number of concentric annuli can reduce the accumulative error only by an unknown extent and results in a loss of computation efficiency.

This paper aims at an accurate and efficient procedure for numerical finite strain solutions of circular tunnels excavated in strain-softening rock masses to provide more practical and applicable design references for soft rock tunnels with high geostress. In the Lagrangian coordinate, the solutions in the elastic, softening and residue region of a circular tunnel are derived in the form of the differential equations. A new procedure for a numerical finite strain solution is implemented based on the iteration and bisection method. The procedure's accuracy and efficiency are verified by comparing it with previous studies. This study performs a global sensitivity analysis of tunnel displacement using this procedure and recommends the supporting stiffness for the design of the Zhongyi tunnel based on the stochastic analysis results.

## 2. Problem Statement

### 2.1. Model Description

The calculation model is shown in Figure 1. It depicts an axisymmetric plane strain issue of a circular tunnel excavated in an infinite, continuous, homogeneous, and isotropic rock mass, subjected to uniform hydrostatic stress $P_0$ at infinity and uniform support pressure $\sigma_0$ exerted on the wall. The original radius $R$ and the deformed radius $r$ represent the initial position and deformed(current) position of a material point, respectively. Consequently, the radial displacement of the material point $u$ can be defined as:

$$u = R - r \tag{1}$$

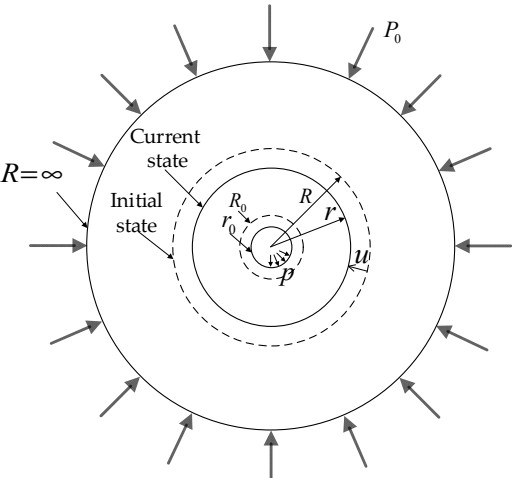

**Figure 1.** Tunnel excavation calculation model based on finite strain theory.

After the excavation, the tunnel radius decreases from its initial value $R_0$ to the deformed radius $r_0$. For a rock mass following elastic-strain-softening behaviour, the rock mass within a deformed radius $r_r$ reaches the residual state, the region outside a deformed radius $r_p$ stays in the elastic state and the rock mass between $r_s$ and $r_p$ is in the strain-softening state.

### 2.2. Governing Equations

For the finite strain model, the stress equilibrium equation of the rock masses in the deformed state can be expressed as the differential equation, Equation (2).

$$\frac{\mathrm{d}\sigma_r}{\mathrm{d}r} + \frac{\sigma_r - \sigma_\theta}{r} = 0 \tag{2}$$

where $\sigma_r$ and $\sigma_\theta$ denote the radial stress and tangential stress, respectively.

The logarithmic function can accurately describe the geometric relationship between radial displacement and strains. In the current polar coordinates, the geometric equation can be expressed as follows:

$$\varepsilon_\theta = \ln\left(\frac{u}{r} + 1\right) \tag{3}$$

$$\varepsilon_r = \ln\left(\frac{du}{dr} + 1\right) \tag{4}$$

where $\varepsilon_\theta$ and $\varepsilon_r$ denote the radial and tangential strains, respectively.

The total strain induced by excavation can be decomposed into the elastic part and the plastic part as follows:

$$\varepsilon_r = \frac{1}{2G}[(1-v)(\sigma_r - P_0) - v(\sigma_\theta - P_0)] + \varepsilon_r^P \tag{5}$$

$$\varepsilon_\theta = \frac{1}{2G}[(1-v)(\sigma_\theta - P_0) - v(\sigma_r - P_0)] + \varepsilon_\theta^P \tag{6}$$

where the shear modulus $G = E/2/(1+v)$, $E$ and $v$ are Young's modulus and Poisson's ratio, respectively; $\varepsilon_r^P = 0$ and $\varepsilon_\theta^P = 0$ in the elastic rock mass.

### 2.3. Softening Parameters, Yield Criterion, and Evolutional Law

The plastic shear strain $\gamma^P$ is the most widely used softening index governing material parameters, which is defined as Equation (7).

$$\gamma^P = \varepsilon_\theta^P - \varepsilon_r^P \tag{7}$$

According to the non-associated MC potential function, the relation of radial and tangential plastic strains can be expressed as Equation (8).

$$\varepsilon_r^P + K(\gamma^P)\varepsilon_\theta^P = 0 \tag{8}$$

where $K(\gamma^P) = \frac{1+\sin\psi(\gamma^P)}{1-\sin\psi(\gamma^P)}$ and $\psi$ is the dilation angle.

The failure of the rock mass is governed by the strain-softening Mohr–Coulomb yielding a function written as:

$$\sigma_\theta = k(\gamma^P)\sigma_r + \sigma_c(\gamma^P) \tag{9}$$

where $k(\gamma^P) = \frac{1+\sin\varphi(\gamma^P)}{1-\sin\varphi(\gamma^P)}$, $\sigma_c(\gamma^P) = \frac{2c(\gamma^P)\cos\varphi(\gamma^P)}{1-\sin\varphi(\gamma^P)}$; $c$ and $\varphi$ are the cohesive strength and the friction angle respectively.

The evolution of material property parameters can be described by the piecewise linear function [38]. Any strength parameter of the rock mass is linearly decreased from the peak value $\eta_P$ to the corresponding residual value $\eta_r$ as shown in Equation (10).

$$\eta(\gamma^P) = \begin{cases} \eta_P & \gamma^P \le 0 \\ \eta_P - (\eta_P - \eta_r)\gamma^P/\gamma^{P*} & 0 < \gamma^P < \gamma^{P*} \\ \eta_r & \gamma^P \ge \gamma^{P*} \end{cases} \tag{10}$$

where $\eta$ denotes any one of the strength parameters such as $\psi$, $c$, and $\varphi$ etc.; $\gamma^{P*}$ is the critical plastic shear strain, the subscripts 'p' and 'r' of $\eta$ denote the peak and residual values respectively.

*2.4. Boundary Conditions*

In the original or deformed state, the excavation disturbance for the rock mass at infinity is infinitely small. Hence, the stress boundary conditions of the rock mass can be expressed in the original coordinate as follows:

$$\begin{cases} \sigma_r = \sigma_\theta = P_0 & R \to \infty \\ \sigma_r = p & R = R_0 \end{cases} \tag{11}$$

**3. Solutions for Stress and Displacement of Rock Mass**

Using Equations (1), (3), and (4), the differential relation between the original coordinate and the deformed is expressed as:

$$\frac{\mathrm{d}r}{r} = \frac{\mathrm{d}R}{R}e^{\varepsilon_\theta - \varepsilon_r} \tag{12}$$

Deriving from the above equations, the stress and displacement solutions are expressed in the form of differential equations of the radial stress $\sigma_r$, the plastic shear strain $\gamma^P$, and the tangential stress $\sigma_\theta$.

*3.1. Solution in the Residue Region*

In the residue region, Equations (7)–(9) are simplified as Equations (13)–(15).

$$\varepsilon_\theta^P = \frac{(1-\sin\psi_r)\gamma^P}{2} \tag{13}$$

$$\varepsilon_r^P + \frac{1+\sin\psi_r}{1-\sin\psi_r}\varepsilon_\theta^P = 0 \tag{14}$$

$$\sigma_\theta = \frac{1-\sin\varphi_r}{1+\sin\varphi_r}\sigma_r + \frac{2c_r\cos\varphi_r}{1-\sin\varphi_r} \tag{15}$$

Using Equations (5), (6), and (9), the differential formula $\mathrm{d}r/r$ can be rewritten as:

$$\frac{\mathrm{d}r}{r} = \frac{\mathrm{d}R}{R} \exp\left[\frac{1}{2G}\left(\frac{-2\sin\varphi_{\mathrm{r}}}{1+\sin\varphi_{\mathrm{r}}}\sigma_{\mathrm{r}} + \frac{2c_{\mathrm{r}}\cos\varphi_{\mathrm{r}}}{1-\sin\varphi_{\mathrm{r}}}\right) + \gamma^{\mathrm{P}}\right] \tag{16}$$

By substituting radial stress and $\mathrm{d}r/r$ in Equation (16) into Equation (2), the first differential equation of the stress equilibrium equation can be obtained, expressed as Equation (17).

$$\frac{\mathrm{d}\sigma_{\mathrm{r}}}{\mathrm{d}R} = \frac{1}{R}\left(\frac{-2\sin\varphi_{\mathrm{r}}}{1+\sin\varphi_{\mathrm{r}}}\sigma_{\mathrm{r}} + \frac{2c_{\mathrm{r}}\cos\varphi_{\mathrm{r}}}{1-\sin\varphi_{\mathrm{r}}}\right)\exp\left[\frac{1}{2G}\left(\frac{-2\sin\varphi_{\mathrm{r}}}{1+\sin\varphi_{\mathrm{r}}}\sigma_{\mathrm{r}} + \frac{2c_{\mathrm{r}}\cos\varphi_{\mathrm{r}}}{1-\sin\varphi_{\mathrm{r}}}\right) + \gamma^{\mathrm{P}}\right] \tag{17}$$

The differential form of the tangential strain can be derived as Equation (18) by differentiating the tangential strain regarding the initial radius in Equation (3) and substituting it into Equation (16).

$$\frac{\mathrm{d}\varepsilon_{\theta}}{\mathrm{d}R} = \frac{1}{R} - \frac{\mathrm{d}r}{r\mathrm{d}R} = \frac{1}{R}\left\{1 - \exp\left[\frac{1}{2G}\left(\frac{-2\sin\varphi_{\mathrm{r}}}{1+\sin\varphi_{\mathrm{r}}}\sigma_r + \frac{2c_{\mathrm{r}}\cos\varphi_{\mathrm{r}}}{1-\sin\varphi_{\mathrm{r}}}\right) + \gamma^{\mathrm{P}}\right]\right\} \tag{18}$$

Seeking the derivative of each term in Equation (6) regarding the initial radius $r$, and another partial differential of the tangential strain can be expressed as Equation (19).

$$\frac{\mathrm{d}\varepsilon_{\theta}}{\mathrm{d}R} = \frac{1}{2G}\left[(1-v)\frac{1-\sin\varphi_{\mathrm{r}}}{1+\sin\varphi_{\mathrm{r}}} - v\right]\frac{\mathrm{d}\sigma_{\mathrm{r}}}{\mathrm{d}R} + \frac{1-\sin\psi_{\mathrm{r}}}{2}\frac{\mathrm{d}\gamma^{\mathrm{P}}}{\mathrm{d}R} \tag{19}$$

From Equations (18) and (19), the second differential equation for the plastic shear strain regarding the initial radius is as follows:

$$\frac{\mathrm{d}\gamma^{\mathrm{P}}}{\mathrm{d}R} = \frac{k_1}{G}\left(v - (1-v)\frac{1}{k}\right)\frac{\mathrm{d}\sigma_{\mathrm{r}}}{\mathrm{d}R} + \frac{2k_1}{R}\left\{1 - \exp\left[\frac{1}{2G}\left(\frac{-2\sin\varphi_{\mathrm{r}}}{1+\sin\varphi_{\mathrm{r}}}\sigma_{\mathrm{r}} + \sigma_{\mathrm{c}}\right) + \gamma^{\mathrm{P}}\right]\right\} \tag{20}$$

where $k_1 = 1/(1-\sin\psi_{\mathrm{r}})$.

The tangential stress can be easily obtained by substituting the radial stress into Equation (7) and the displacement of rock mass material point can be calculated using Equations (1), (3), (6), and (13).

$$u = R\left\{1 - \exp\left[\frac{-(1-v)(\sigma_{\mathrm{r}} - P_0) + v(\sigma_{\theta} - P_0)}{2G} - \frac{(1-\sin\psi_{\mathrm{r}})\gamma^{\mathrm{P}}}{2}\right]\right\} \tag{21}$$

### 3.2. Solution in the Softening Region

The derivation method of two differential equations for the radial stress and the plastic shear strain in the softening region is like that in the residue region. Because the rock mass strength properties vary with the plastic shear strain, the differential equations become more complex.

Using Equations (5)–(7) and (9), the differential formula $\mathrm{d}r/r$ is rewritten as follows:

$$\frac{\mathrm{d}r}{r} = \frac{\mathrm{d}R}{R} \exp\left\{\frac{1}{2G}\left[(k(\gamma^{\mathrm{P}}) - 1)\sigma_{\mathrm{r}} + \sigma_{\mathrm{c}}(\gamma^{\mathrm{P}})\right] + \gamma^{\mathrm{P}}\right\} \tag{22}$$

Substituting the tangential stress in Equation (7) and $\mathrm{d}r/r$ in Equation (22) into the stress equilibrium equation, the first differential equation of the radial stress regarding the original radius is obtained as follows:

$$\frac{\mathrm{d}\sigma_{\mathrm{r}}}{\mathrm{d}R} = \frac{(k(\gamma^{\mathrm{P}}) - 1)\sigma_{\mathrm{r}} + \sigma_{\mathrm{c}}(\gamma^{\mathrm{P}})}{R}\exp\left\{\frac{1}{2G}\left[(k_{\mathrm{P}}(\gamma^{\mathrm{P}}) - 1)\sigma_{\mathrm{r}} + \sigma_{\mathrm{c}}(\gamma^{\mathrm{P}})\right] + \gamma^{\mathrm{P}}\right\} \tag{23}$$

Like Equations (18) and (19), the differential form of the tangential strain to the original radius can be written as Equations (24) and (25) respectively.

$$\frac{d\varepsilon_\theta}{dR} = \frac{1}{R} - \frac{1}{R}\exp\left[\frac{(k_p(\gamma^P)-1)\sigma_r + \sigma_c(\gamma^P)}{2G} + \gamma^P\right] \tag{24}$$

$$\frac{d\varepsilon_\theta}{dR} = \frac{1}{2G}\left[(1-v)\frac{d\sigma_\theta}{dR} - v\frac{d\sigma_r}{dR}\right] + \frac{d\varepsilon_\theta^P}{dR} \tag{25}$$

where $\frac{d\sigma_\theta}{dR} = \left[\sigma_r\frac{\partial k(\gamma^P)}{\partial\gamma^P} + \frac{\partial\sigma_c(\gamma^P)}{\partial\gamma^P}\right]\frac{d\gamma^P}{dR} + k(\gamma^P)\frac{d\sigma_r}{dR}$ and $\frac{d\varepsilon_\theta^P}{dR} = \frac{\partial\left[\frac{\gamma^P}{1+K(\gamma^P)}\right]}{\partial\gamma^P}\frac{d\gamma^P}{dR}$.

Hence, the differential equation for the plastic shear strain in the softening region is derived as Equation (26).

$$\frac{d\gamma^P}{dR} = \frac{\frac{1}{R} - \frac{1}{R}exp\left[\frac{(k(\gamma^P)-1)\sigma_r+\sigma_c(\gamma^P)}{2G} + \gamma^P\right] - \frac{(1-v)k(\gamma^P)-v}{2G}\frac{d\sigma_r}{dR}}{\frac{1-v}{2G}\left[\sigma_r\frac{\partial k(\gamma^P)}{\partial\gamma^P} + \frac{\partial\sigma_c(\gamma^P)}{\partial\gamma^P}\right] + \frac{\partial\left[\frac{\gamma^P}{1+K(\gamma^P)}\right]}{\partial\gamma^P}} \tag{26}$$

After solving Equations (23) and (26), the tangential stress can be obtained easily by substituting the radial stress into Equation (7) and the displacement of the rock mass material point in the softening region can be calculated by Equation (27).

$$u = R\left\{1 - \exp\left[\frac{(vk(\gamma^p)-(1-v))\sigma_r + (1-2v)P_0 + v\sigma_c(\gamma^P)}{2G} - \frac{(1-\sin\psi(\gamma^P))\gamma^P}{2}\right]\right\} \tag{27}$$

### 3.3. Solution in the Elastic Region

In the elastic region, there is no plasticity. Hence, the radial stress and tangential stress are taken as basic variables. The stress equilibrium equation and the compatibility equation can be easily rewritten into differential equations of the radial stress and the tangential stress, as Equations (28) and (29).

$$\frac{d\sigma_r}{dR} + \frac{\sigma_r - \sigma_\theta}{R}\exp\left(\frac{\sigma_\theta - \sigma_r}{2G}\right) = 0 \tag{28}$$

$$\frac{d\sigma_\theta}{dR} = \frac{1}{1-v}\left\{\frac{2G}{R}\left[1 - \exp\left(\frac{\sigma_\theta - \sigma_r}{2G}\right)\right] + v\frac{\sigma_\theta - \sigma_r}{R}\exp\left(\frac{\sigma_\theta - \sigma_r}{2G}\right)\right\} \tag{29}$$

The displacement of the rock mass material point in the elastic region is calculated using Equation (30).

$$u = R\left\{1 - \exp\left[\frac{-(1-v)(\sigma_r - P_0) + v(\sigma_\theta - P_0)}{2G}\right]\right\} \tag{30}$$

## 4. Numerical Implementation Procedure

According to Equations (28)–(30), the stress and displacement of the rock mass in the elastic region are determined by the initial values of the radial stress and tangential stress on the boundary between the elastic region and the plastic region ($R_p$). In the softening region, the tangential stress can be calculated by the radial stress and the plastic shear strain by the yielding function. Accordingly, the radial stress and tangential stress on $R_p$ can be obtained by solving Equations (23) and (26), once the initial values of the radial stress and the plastic shear strain on the boundary between the softening region and the residue region ($R_r$) are known. Similarly, the initial values of the radial stress and the plastic shear strain on the tunnel inner boundary ($R_0$) determine the stress, the plastic shear strain, and the displacement of any point in the residue region according to Equations (17), (20), and (21). Hence, there are strict one-to-one correspondences between the rock mass stress $P_0$

at infinity and the plastic shear strain which decrease monotonously in the residue and softening regions with $R$ increasing when the support pressure is given.

### 4.1. Numerical Procedures for Calculating Limit Support Pressure

When the support pressure equals the softening limit value $p_{rp}^*$, the plastic shear strain on the tunnel inner boundary $R_0$ exactly equals the critical plastic shear strain $\gamma^{p*}$. Comparatively, the plastic shear strain is equivalent to zero and the tangential stress and radial stress on the tunnel inner boundary $R_0$ just meet the yield criteria when the support pressure equals the elastic limit value $p_{pe}^*$. There are three states of the surrounding rock, the residue-softening-elastic state, the softening-elastic state, and the elastic state, with the support pressure being less than $p_{rp}^*$, between $p_{rp}^*$ and $p_{pe}^*$ and greater than $p_{pe}^*$ respectively. The flowcharts depicting the calculation procedures of $p_{rp}^*$ and $p_{pe}^*$ are shown in Figures 2 and 3, respectively.

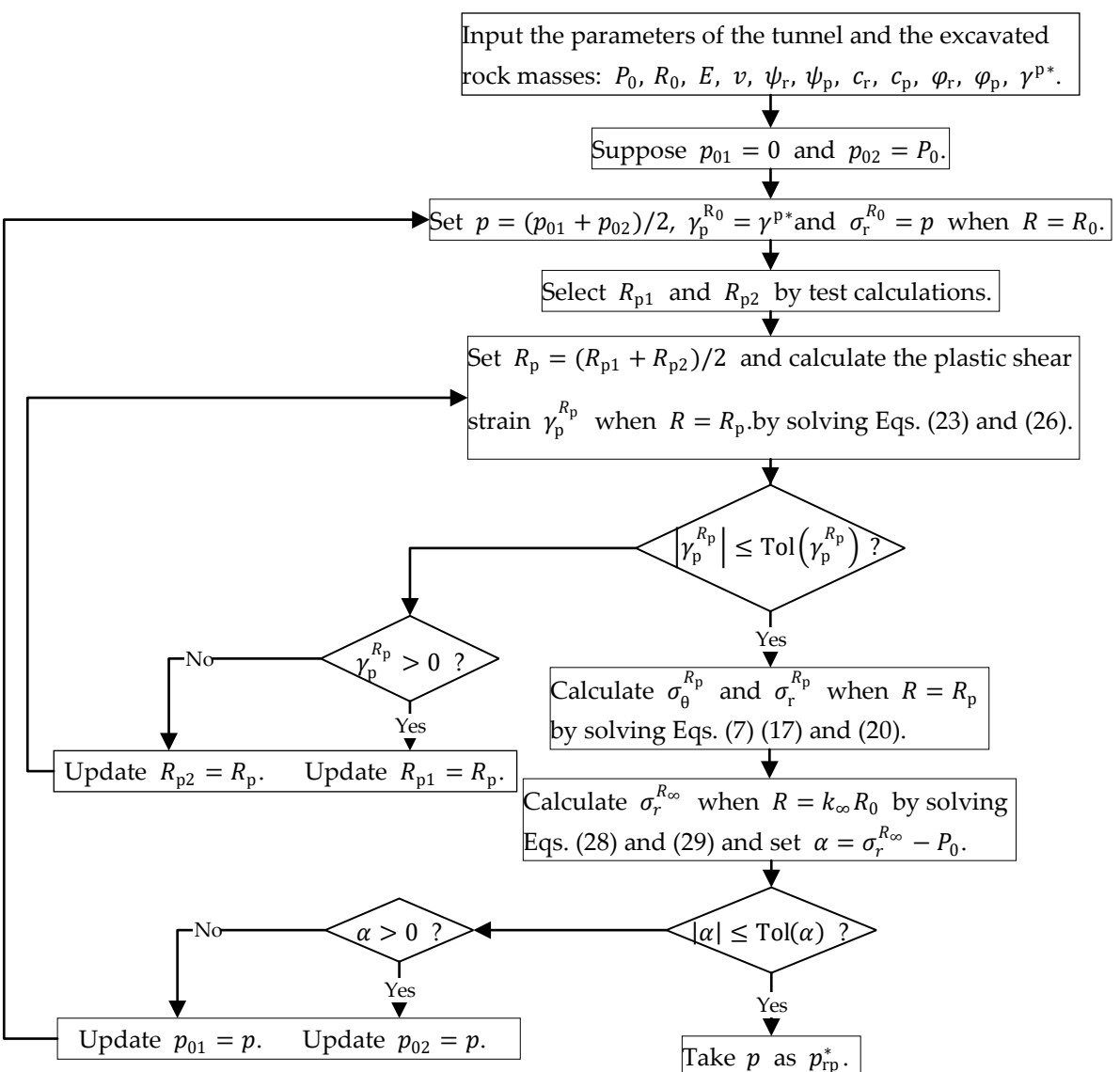

**Figure 2.** Calculation flowchart of $p_{rp}^*$.

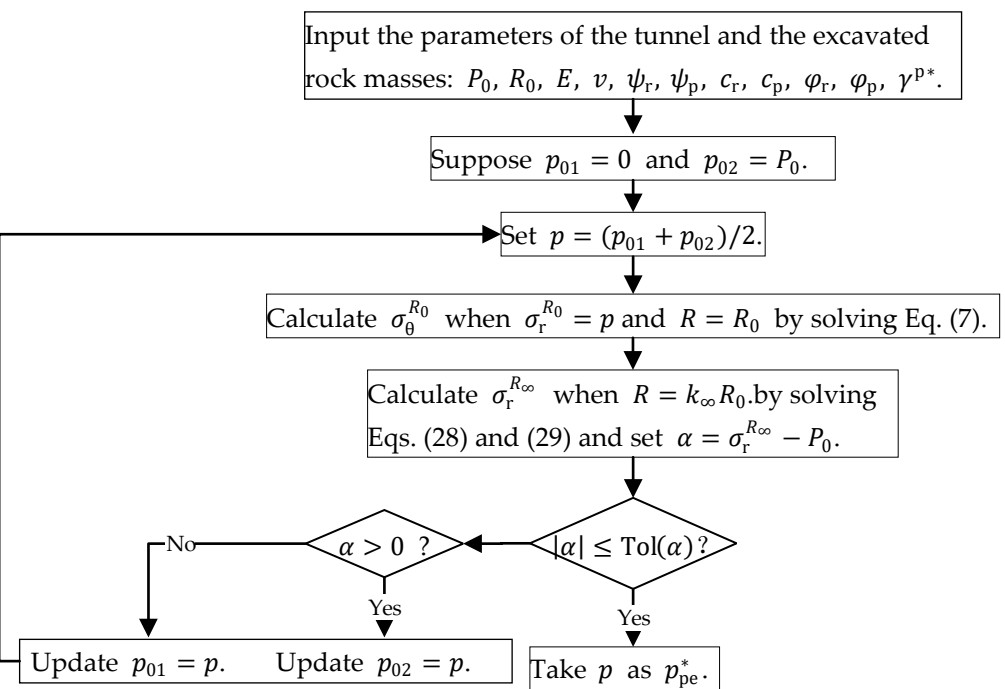

**Figure 3.** Calculation flowchart of $p_{pe}^*$.

The error tolerances of $\alpha$ and $\gamma_p^{R_p}$ and the value of $k_\infty$ have an important influence on the precision and accuracy of the calculation results. Ordinarily, greater $k_\infty$ and smaller $\text{Tol}\left(\gamma_p^{R_p}\right)$ and $\text{Tol}(\alpha)$ lead to better precision and higher accuracy of the results. In this paper, $\text{Tol}\left(\gamma_p^{R_p}\right)$, $\text{Tol}(\alpha)$, and $k_\infty$ are taken as to $10^{-11}$, $10^{-8}$, and $10^5$ respectively, of which the rationality is verified in Section 5.

In Figure 2, the first values of $R_{p1}$ and $R_{p2}$ in the loop can be determined by the following method: (1) set $R_p(k) = R_0 + \delta_1 + (k-1)\delta_2$ and calculate the plastic shear strain $\gamma_p^{R_p}(k)$ when $R = R_p(k)$ by solving Equations (17) and (20); (2) repeat the above calculation from $k = 1$ until $\gamma_p^{R_p}(k) > 0$ and $\gamma_p^{R_p}(k+1) < 0$ and take $R_p(k)$ and $R_p(k+1)$ as $R_{p1}$ and $R_{p2}$ respectively. The value of $\delta_1$ and $\delta_2$ can be adjusted according to the accuracy and computational efficiency. For the stability of solutions, take $\delta_1$ and $\delta_2$ equal to $10^{-5}R_0$ and $10^{-4}R_0$, respectively, in this paper.

### 4.2. Numerical Procedures for Three States

On the condition that the support pressure and other tunnel parameters are given, the stress and displacement calculation can be divided into three scenarios when the support pressure $p$ is between $p_{pe}^*$ and $P_0$, between $p_{pe}^*$ and $P_0$, and smaller than $p_{pe}^*$. The following sections elaborate the numerical procedures for the calculation of stress and displacement in the above three states.

#### 4.2.1. Numerical Procedure for Elastic State

When the support pressure is greater than $p_{pe}^*$ and smaller than the geostress $P_0$, the rock mass stays elastic and the stress and displacement of the rock masses can be obtained by the procedure coded as the flowchart in Figure 4.

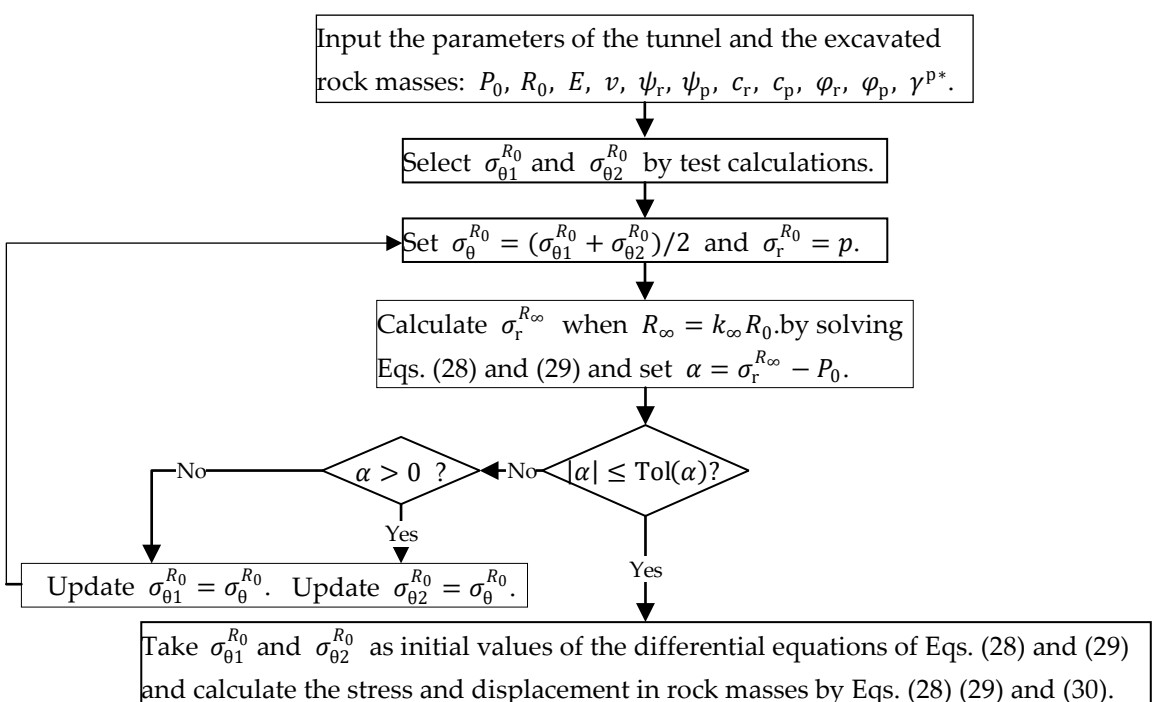

**Figure 4.** Calculation flowchart for elastic state.

In Figure 4, the selecting method of the first values of $\sigma_{\theta 1}^{R_0}$ and $\sigma_{\theta 2}^{R_0}$ is like that of $R_{p1}$ and $R_{p2}$ in Section 3. At first, calculate $\sigma_r^{R_\infty}$ when $R = k_\infty R_0$.and $\sigma_\theta^{R_0}(k) = 0.1(k-1)P_0$ by solving Equations (28) and (29) and repeat this calculation from $k = 1$ until $\sigma_r^{R_\infty}(k) > P_0$ and $\sigma_r^{R_\infty}(k+1) < P_0$. Then take $\sigma_r^{R_\infty}(k)$ and $\sigma_r^{R_\infty}(k+1)$ as the initial values of $\sigma_{\theta 1}^{R_0}$ and $\sigma_{\theta 2}^{R_0}$ in the loop respectively.

### 4.2.2. Numerical Procedure for Softening-Elastic State

There are the softening region and the elastic region in the rock masses when the support pressure $p$ is between $p_{rp}^*$ and $p_{pe}^*$. The calculation for stress and displacement is implemented as the flowchart in Figure 5.

In Figure 5, the selecting method of the first values of $R_{p1}$ and $R_{p2}$ is the same as that in Figure 3. In order to get the first values of $\gamma_{p1}^{R_0}$ and $\gamma_{p2}^{R_0}$, calculate $\alpha(k)$ when $\gamma_p^{R_0}(k) = \delta_3 + (k-1)\delta_4$ and repeat this calculation from $k = 1$ until the signs of $\alpha(k)$ and $\alpha(k+1)$ are contrary. Then take $\gamma_p^{R_0}(k)$ and $\gamma_p^{R_0}(k+1)$ as the initial values of $\gamma_{p1}^{R_0}$ and $\gamma_{p2}^{R_0}$ in the cycle computation. The value of $\delta_3$ and $\delta_4$ can be taken as $10^{-6}\gamma^{p*}$ and $10^{-2}\gamma^{p*}$ respectively, or smaller.

### 4.2.3. Numerical Procedure for Elastic-Softening-Residue State

The numerical procedure for the elastic-softening-residue state shown in Figure 6 adds the searching procedure for the radius of the residue region $R_r$ by the bisection method, compared with the procedure in Figure 5. Once the initial value of the plastic shear strain on the position $R_0$ and the radii of the residue region and the plastic region are approximated adequately accurately by the method of bisection while the other initial value of the radial stress equals the support pressure, the stress and displacement of any point in the rock masses can be calculated by solving Equations (17), (20), (21), (23), and (26)–(30).

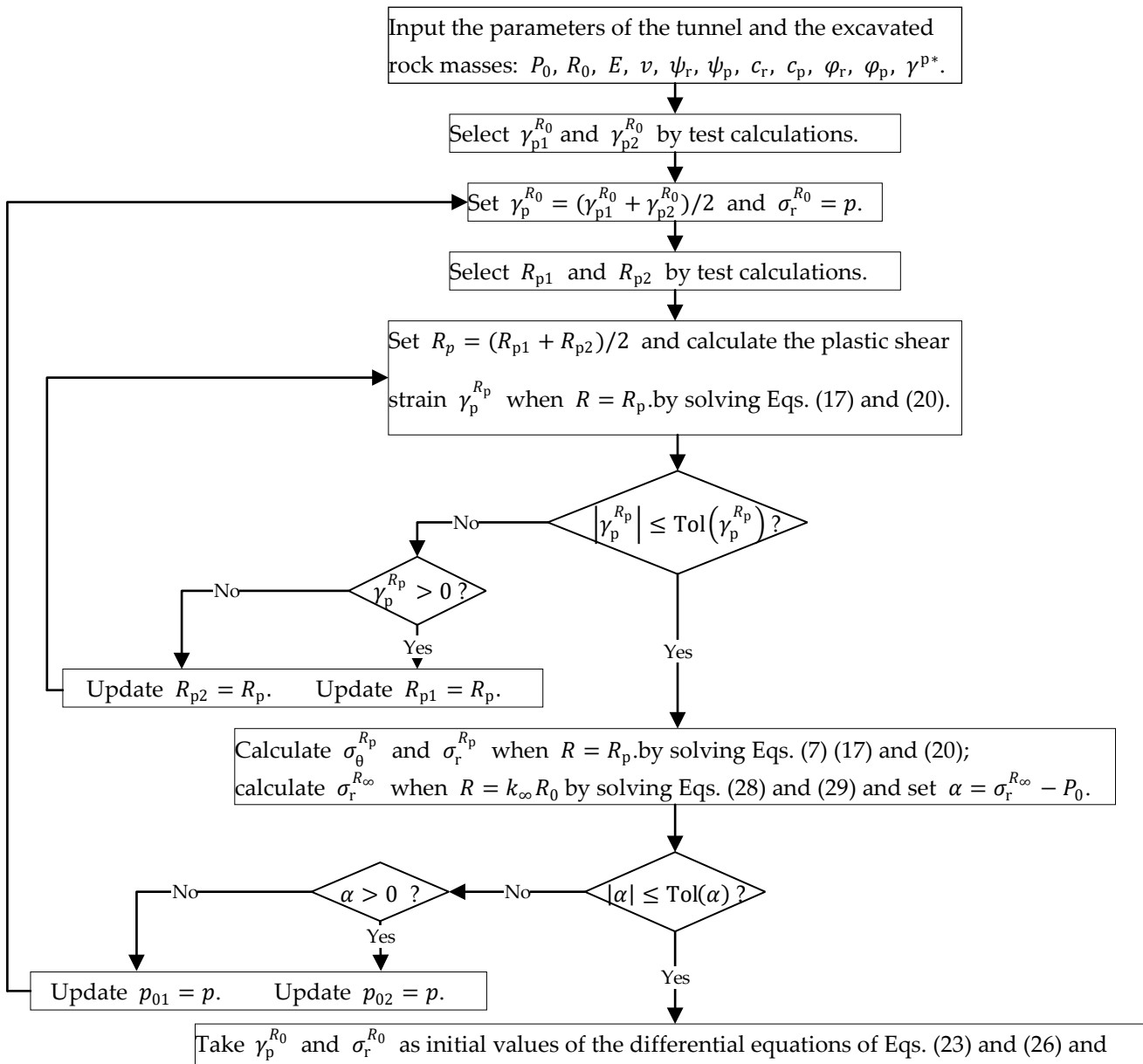

**Figure 5.** Calculation flowchart for softening-elastic state.

In Figure 6., the searching process for the first values of $R_{r1}$ and $R_{r2}$ is like the above selecting process. Similarly, calculate iteratively the plastic shear strain $\gamma_p^{R_r}(k)$ when $R = R_r(k) = R_0 + \delta_1 + (k-1)\delta_2$ from $k = 1$ until $\gamma_p^{R_r}(k) > \gamma^{p*}$ and $\gamma_p^{R_r}(k+1) < \gamma^{p*}$ and then take $R_r(k)$ and $R_r(k+1)$ as the initial values.

The above five numerical procedures are integrated into the finite strain procedure.

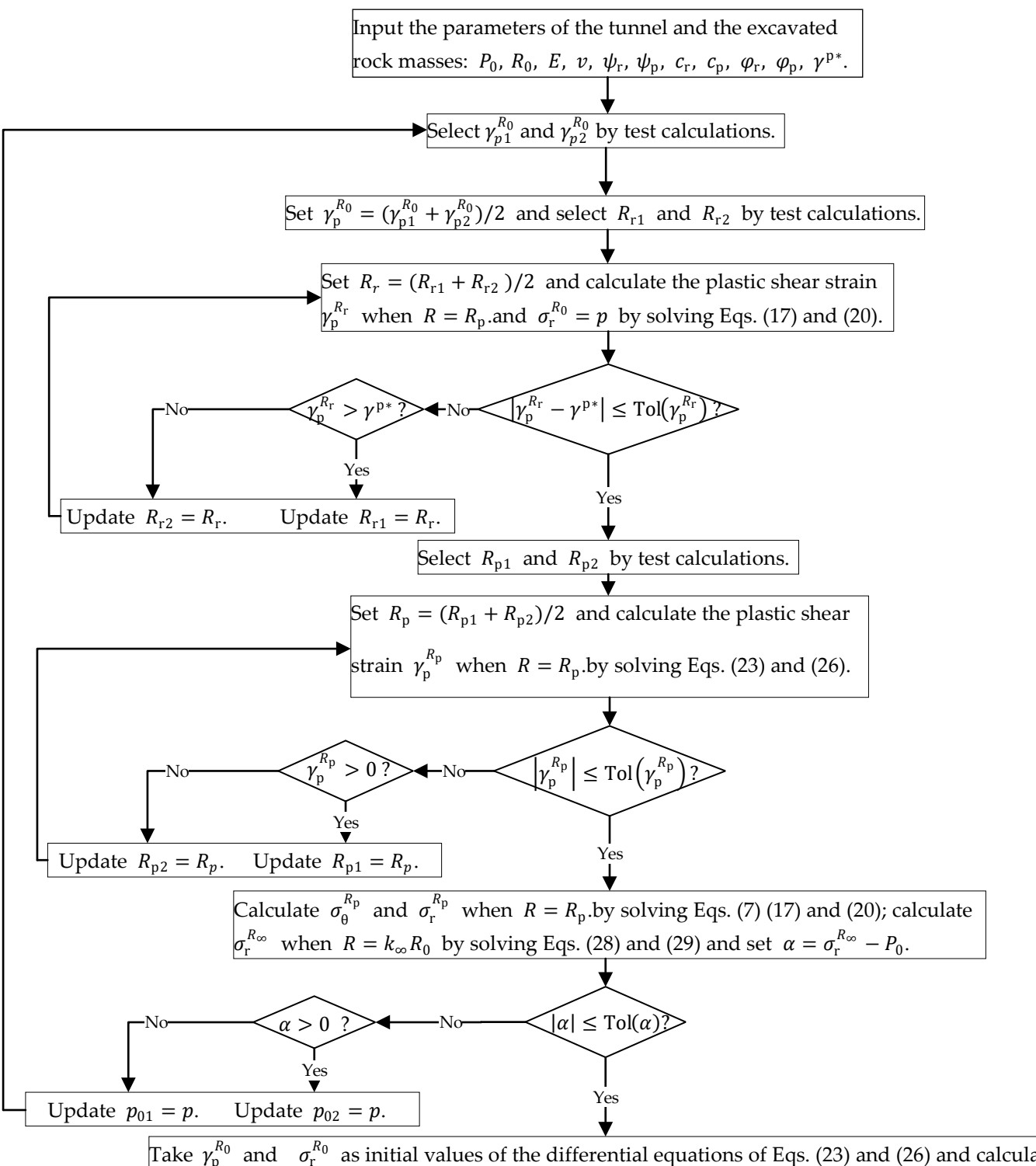

**Figure 6.** Calculation flowchart for elastic-softening-residue state.

## 5. Verification, Comparison, and Discussion

### 5.1. Verification for Finite Strain Procedure

For the verification, this study compares the proposed solution with the numerical solutions in [37,38]. The parameters of the MC rock mass listed in Table 1 are the same as those in 'Figure 6' provided by Ref. [37]. The relations between the dimensionless

displacement $u_0/R_0$, $R_p/R_0$, $R_r/R_0$ and the dimensionless supporting pressure $p/P_0$ are visualized by the scattered diagrams in Figure 7 with two sets of data calculated by the procedure in this paper and former studies. Ref. [37] presents the results solved by the recursive program (Zhang's solution) and obtained by the finite finite-difference method (FDM).

**Table 1.** Geometry and material property parameters.

| MC Rock | Value |
|---|---|
| Radius of opening, $R_0$ (m) | 3 |
| Initial stress, $P_0$ (MPa) | 1 |
| Young's modulus, $E$ (MPa) | 30 |
| Poisson ratio, $v$ (-) | 0.3 |
| Critical plastic shear strain, $\gamma^{p*}$ (-) | 0.15 |
| $c_p$ (MPa) | 0.2 |
| $c_r$ (MPa) | 0.02 |
| $\varphi_p$ (Deg) | 40 |
| $\varphi_r$ (Deg) | 20 |
| $\psi_p$ (Deg) | 10 |
| $\psi_r$ (Deg) | 5 |

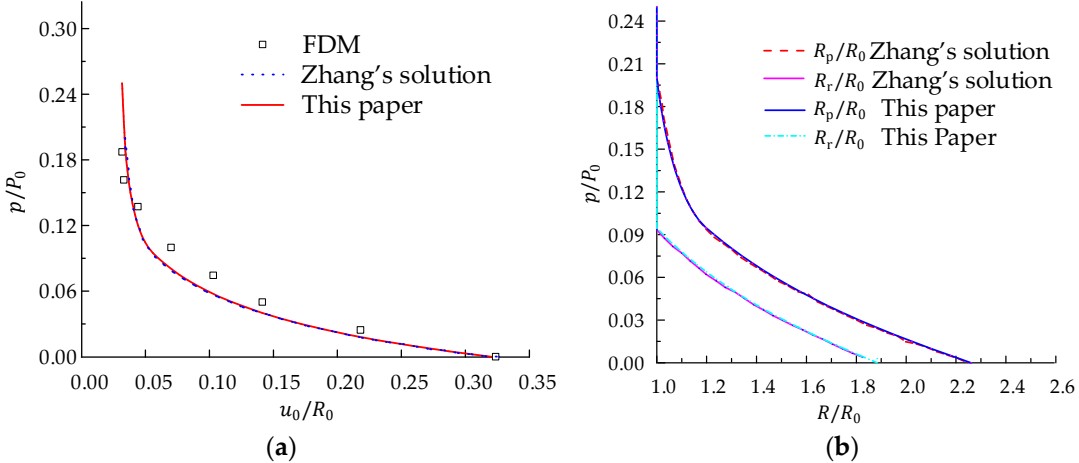

**Figure 7.** Comparison between the results of this paper and Zhang's: (**a**) Ground response curve (GRC); (**b**) radii of residue region and softening region.

By the proposed procedure in this paper, the calculated limit support pressures $p_{rp}^*$ and $p_{pe}^*$ are equal to 0.0945 MPa and 0.2006 MPa respectively. As shown in Figure 7, the GRC and the evolutions of the softening region radius and residual region radius calculated by the proposed procedure are virtually identical with those of Zhang's solution, which verifies the accuracy of the proposed numerical finite strain procedure. Without the supporting pressure, $u_0/R_0$, $R_p/R_0$, and $R_r/R_0$ are 0.3196, 1.884, and 2.260 respectively, slightly different with 0.3235, 1.87 and 2.25 in Zhang's study. This difference is caused mainly by the approximant treatment of the infinity.

In Zhang's study, $k_\infty$ is taken as 50 to approximate the infinite boundary, whereas $k_\infty$ equals $10^5$ in this section. As shown in Figure 8, the value of $k_\infty$ obviously affects the computational accuracy of $u_0/R_0$, $R_p/R_0$ and $R_r/R_0$. When $k_\infty$ reaches 1000, the calculation results stabilize. In addition, the iteration times $n$ have a decisive influence on the accuracy as well. As shown in Figure 9, with the number of iterations increasing, the values of $u_0/R_0$, $R_p/R_0$ and $R_r/R_0$ gradually converge and the magnitude of the calculation error $|\alpha|$ decreases to about $10^{-14}$ which reaches the highest accuracy when $k_\infty = 10^7$. After iterating only 10 times, $u_0/R_0$, $R_p/R_0$ and $R_r/R_0$ become rather stable. When the tolerances of the calculation relative error Tol($\alpha$) are $10^{-6}$, $10^{-8}$, $10^{-10}$, $10^{-12}$,

and $10^{-14}$, the time for one whole calculation is 0.89 s, 1.18 s, 1.72 s, 1.98 s, and 2.82 s in this numerical example. It shows that for the proposed procedure the significant improvement of the precision does not affect the calculation efficiency excessively.

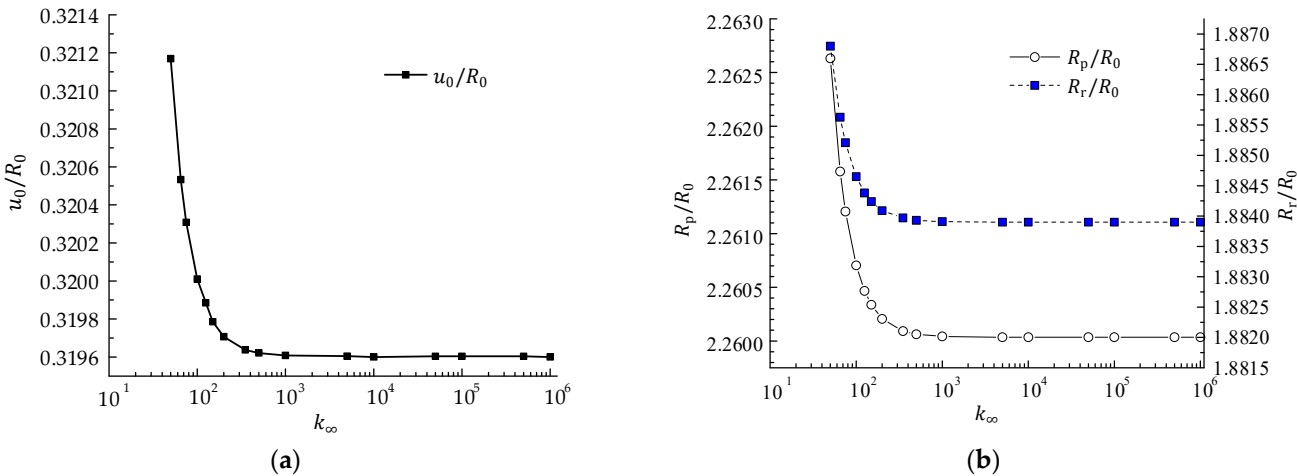

**Figure 8.** Influence of $k_\infty$: (**a**) $u_0/R_0$ varying with $k_\infty$; (**b**) plastic radii varying with $k_\infty$.

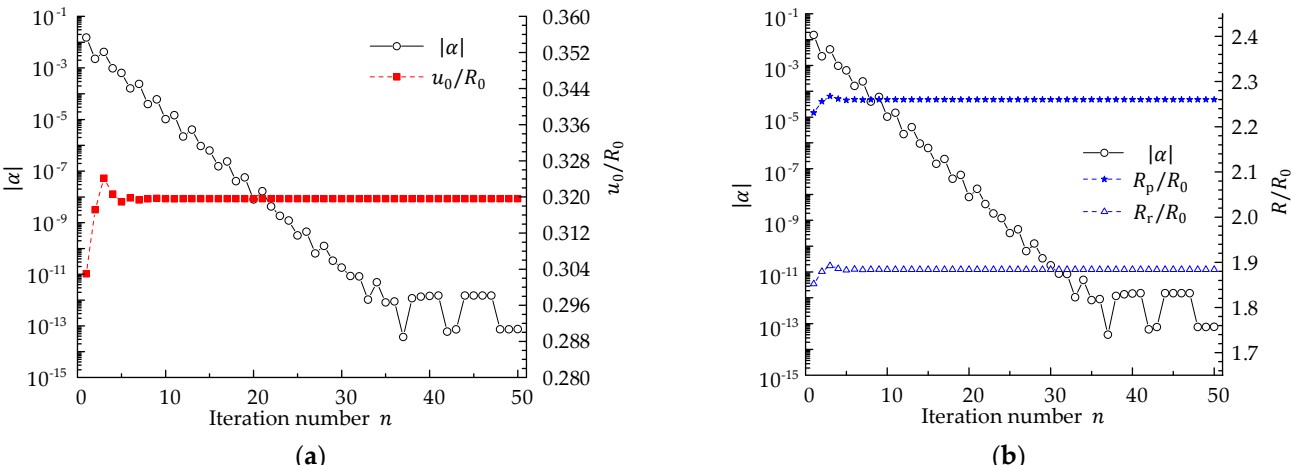

**Figure 9.** Influence of iteration times: (**a**) $u_0/R_0$ and $|\alpha|$ varying with $n$; (**b**) plastic radii and $|\alpha|$ varying with $n$.

*5.2. Comparison for Small Strain Procedure*

Likewise, this iteration method can be applied to solve the excavating problem in strain-softening rock masses under the small deformation hypothesis. The differential equations and the calculation formula for the displacement in the elastic region, the softening region, and the residue region are derived and listed in Appendix A. Following the procedures in Figures 2–6, the numerical procedure for circular tunnels excavated in strain-softening rock masses is programmed based on the small deformation hypothesis.

In Figure 10, solutions obtained by finite strain procedure and small strain procedure are compared with the authoritative Alonso's self-similar solution [8], which is based on the small deformation hypothesis. The small strain solution is consistent with Alonso's result, verifying the correctness and strong applicability of this iteration method. The finite strain solutions, $u_0$, $R_p$ and $R_r$, are smaller than the small strain solutions. That the gap between the two solutions decreases with the supporting pressure increasing, illustrates that the small strain hypothesis underestimates the self-bearing capacity of the surrounding rock and the finite deformation theory is more appropriate for the analysis of large deformation tunnels.

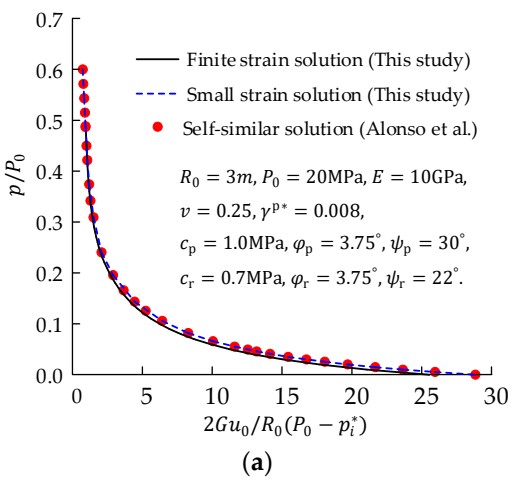
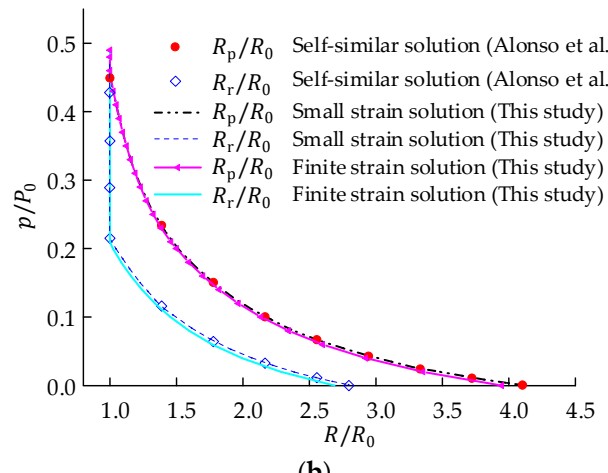

**Figure 10.** Comparison of finite strain solution, small strain solution, and Alonso's self-similar solution: (**a**) $2Gu_0/R_0\left(P_0 - p_i^*\right)$ varying with $p/P_0$; (**b**) plastic radii varying with $p/P_0$.

## 6. Application for Tunnel Design

During the construction of the Zhongyi tunnel in the Yunnan province of China, over 44% of the tunnel encountered large deformation and the maximum deformation was larger than 600 mm. The main reason for this problem was the mismatch between the support stiffness and the weak surrounding rock conditions [37,40]. This section adopts the finite strain procedure to calculate the needed stiffness of the whole supporting system. The three-cantered arch section is equivalent to a circular tunnel with a radius of 4.25 m. The in-situ stress and rock mass parameters are as follows: $P_0 = 12$ MPa, $E = 0.95$ GPa, $v = 0.3$, $\gamma^{p*} = 0.035$, $c_p = 0.48$ MPa, $c_r = 0.25$ MPa, $\varphi_p = 26°$, $\varphi_r = 20°$, $\psi_p = 16°$, $\psi_r = 12°$ [37].

First, the finite strain procedure is integrated into one MATLAB toolbox called SAFE (Sensitivity Analysis For Everybody) [41] to perform the sensitivity analysis of tunnel deformation. The boundary of rock mass parameters is $(\mu - 3\sigma, \mu + 3\sigma)$, where $\mu$ and $\sigma$ denote the mean value and the standard deviation of each parameter. The distribution type of the rock mass parameters is normal distribution [42]. It is supposed that the coefficient of variations (COVs) of the in-situ stress and of other parameters are 0.1 and 0.05, respectively. The support pressure is sampled randomly and uniformly between 0 and $0.25P_0$. According to the global sensitivity indices (Sobol indices) of each parameter in Figure 11, tunnel deformation is most sensitive to the support pressure $p$ compared to other factors except for the inexorable geostress $P_0$. Therefore, improving the support stiffness is the most direct and effective means to control the tunnel deformation.

Furthermore, the stochastic analysis is performed by the Monte Carlo method to optimize the stiffness of the support structure for the Zhongyi tunnel. All distribution parameters are the same as the above but the range of the support pressure is $[0, 0.5P_0]$. The sampling number is 20,000 and the time for each calculation is between 0.9 s and 15 s in a computer with Intel Core *I* 7-6700 CPU. The tunnel displacement and support pressure are scattered in Figure 12a and the upper and lower envelope curves are depicted. As the support pressure increases, the variation range of $u_0$ becomes smaller. When the deformation limit is 0.5 m, the range of the support pressure is between 0.72 MPa and 1.63 MPa.

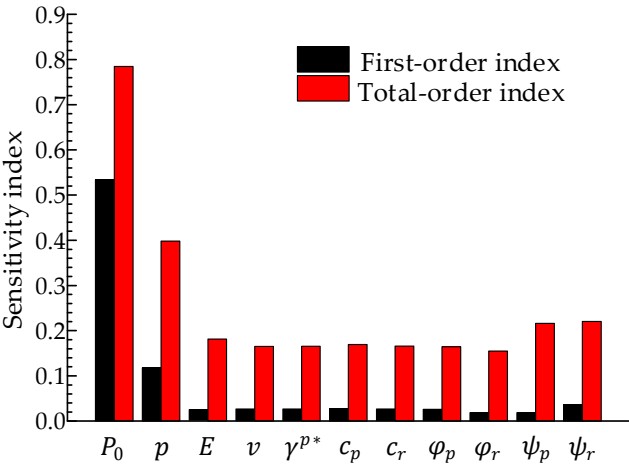

**Figure 11.** Sensitivity index of tunnel deformation.

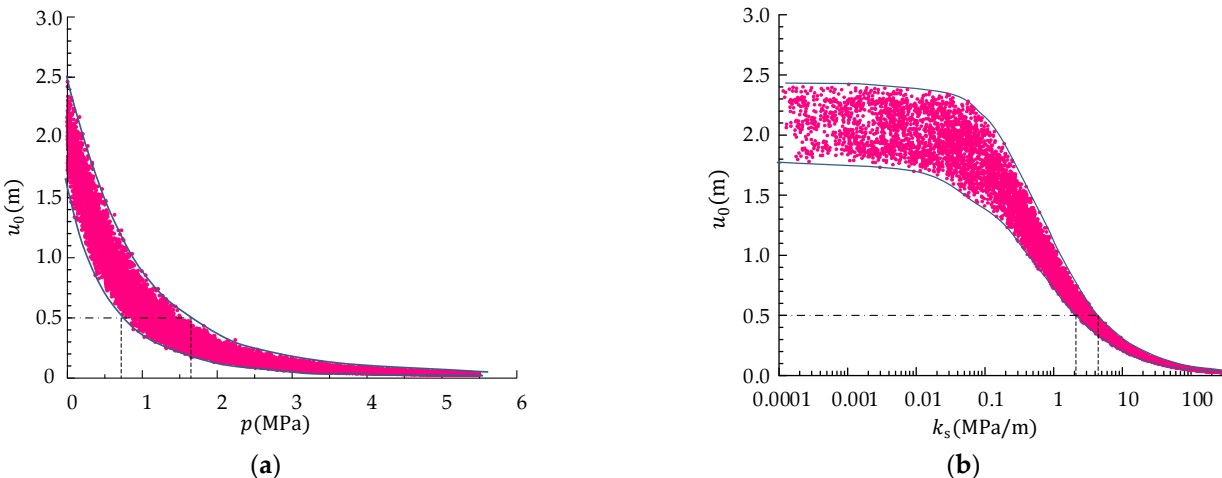

**Figure 12.** Stochastic sampling results: (**a**) tunnel displacement and support pressure; (**b**) tunnel displacement and support stiffness.

In the specific design process for the tunnel, calculating the needed stiffness is more practical than predicting the tunnel displacement. Hence, the combined stiffness of the primary support and the secondary lining is obtained by taking the displacement release coefficient as 0.25 before installing the support structure [43]. As shown in Figure 12b, the discrete degree of the displacement $u_0$ significantly decreases with the stiffness $k$ increasing. When the combined stiffness is larger than 4.3 MPa/m, it can be guaranteed that the tunnel displacement $u_0$ will not exceed 0.5 m, smaller than the limit deformation. As for the detail design, references [44–53] can be used to calculate the support stiffness, optimize the support parameters, analyze the tunnel stability and perform the whole stiffness design for the tunnel-support system.

## 7. Conclusions

This paper proposes a new numerical finite strain procedure for a circular tunnel in MC strain-softening rock masses with a non-associated flow rule. First, the explicit form of the differential relation between the original coordinate and the deformed coordinate is established in the Lagrangian coordinate. Using the governing equations, yield criterion, evolutional law, the solutions in the residue region, and the softening region are derived as two sets of differential equations of radial stress and plastic shear strain and the solutions in the elastic region are expressed as the differential equations of radial stress and tangential stress. By directly solving the differential equations using the Runge–Kutta method, the

problem of solving for the finite strain solution transforms into the problem of searching for two adjacent boundaries of three regions ($R_\mathrm{s}$ and $R_\mathrm{p}$) and the initial values on the excavation boundary ($R_0$).

Then the bisection method is adopted to approximate the true value based on the characteristics that the dependent variables (radial stress, plastic shear strain and tangential stress) in each region increase or decrease monotonously with the deformed radius. Two numerical procedures of calculating $p_\mathrm{rp}^*$ and $p_{pe}^*$ and three procedures for the solutions in the residue-softening-elastic state, the softening-elastic state, and the elastic state respectively are programmed and integrated into the finite strain numerical procedure. The proposed finite strain procedure is validated by comparing with recursive solutions and FLAC simulation results. By increasing $k_\infty$ and $n$, the calculation relative error $|\alpha|$ decreases to about $10^{-14}$. When $k_\infty$ and $n$ are greater than 1000 and 10 respectively for this numerical example, $u_0/R_0$, $R_\mathrm{p}/R_0$ and $R_\mathrm{r}/R_0$ become rather stable. The proposed procedure directly solves three sets of differential equations and uses the bisection method to approximate the boundary conditions, realizing quantitative control for the calculation error. More meaningfully, the highly efficient computing performance of the proposed procedure creates conditions for its engineering application in the global sensitivity analysis and the reliability design of the Zhongyi tunnel, which requires large sample data.

Moreover, this paper develops a numerical procedure for circular tunnels excavated in strain-softening rock masses based on the small deformation hypothesis still using the iteration method. The correctness of this iteration method was verified again by comparing with a self-similar solution. However, this solution underestimates the self-bearing capacity of the surrounding rock and is inappropriate to be used to analyze a large deformation tunnel.

In the end, the finite strain procedure was adopted for the sensitivity analysis and the reliability design of Zhongyi tunnel. The sensitivity indices of tunnel deformation to support pressure and geostress are much higher than others. This study recommends 4.3 MPa/m as the designed support stiffness to guarantee a tunnel displacement lower than 0.5 m.

**Author Contributions:** Conceptualization, W.C. and D.Z.; methodology, W.C. and Q.F.; software, W.C.; validation, W.C.; formal analysis, W.C.; investigation, W.C., X.C. and T.X.; writing—original draft preparation, W.C.; writing—review and editing, D.Z. and Q.F.; visualization, W.C., X.C. and T.X.; supervision, D.Z.; project administration, D.Z. and Q.F.; funding acquisition, D.Z. All authors have read and agreed to the published version of the manuscript.

**Funding:** This research was supported financially by the National Key Research and Development Program of China (Grant No. 2021YFB2300803 and Grant No. 2017YFC0805401) and National Natural Science of China (Grant No. 51738002).

**Institutional Review Board Statement:** Not applicable.

**Informed Consent Statement:** Not applicable.

**Data Availability Statement:** Data are contained within this article.

**Acknowledgments:** The authors are grateful to all editors' and reviewers' work which improved the results and the presentation of this paper. Further, the authors thank Francesca Pianosi for making the MATLAB toolbox SAFE [41] publicly available.

**Conflicts of Interest:** The authors declare no conflict of interest.

### Abbreviations

The following abbreviations are used in this manuscript:

| | |
|----|----|
| MC | Mohr–Coulomb |
| DF | Dimensional factor |
| DA | Divergence angles |

**Appendix A**

With the small deformation hypothesis, the formulas for stress and displacement of rock mass in three regions are derived and listed as follows. All formula symbols are consistent with the above formulas.

(1) Solutions in residue region

The solution of the radial stress and the tangential stress in residue region can be expressed as Equations (A1) and (A2).

$$\sigma_{\mathrm{r}} = \frac{\sigma_0(k_{\mathrm{r}} - 1) + \sigma_{\mathrm{c}}}{k_{\mathrm{r}} - 1}\left(\frac{r}{R_0}\right)^{k_{\mathrm{r}} - 1} - \frac{\sigma_{\mathrm{c}}}{k_{\mathrm{r}} - 1} \tag{A1}$$

$$\sigma_{\theta} = \frac{k_{\mathrm{r}}[\sigma_0(k_{\mathrm{r}} - 1) + \sigma_{\mathrm{c}}]}{k_{\mathrm{r}} - 1}\left(\frac{r}{R_0}\right)^{k_{\mathrm{r}} - 1} - \frac{\sigma_{\mathrm{c}}}{k_{\mathrm{r}} - 1} \tag{A2}$$

where $k_r = \frac{1+\sin\varphi_r}{1-\sin\varphi_r}$ and $\sigma_c(\gamma^p) = \frac{2c_r\cos\varphi_r}{1-\sin\varphi_r}$. The solution of the displacement in the residue region is expressed as Equation (A3).

$$u = C\frac{r^{k_{\mathrm{r}}} - r^{-K}R_0{}^{k_{\mathrm{r}}+K}}{K + k_{\mathrm{r}}} + D\frac{r - r^{-K}R_0{}^{K+1}}{K + 1} + r^{-K}R_0{}^K u_0 \tag{A3}$$

where $C = \frac{1}{2G}\frac{\sigma_0(k_{\mathrm{r}}-1)+\sigma_c}{(k_{\mathrm{r}}-1)R_0{}^{k_{\mathrm{r}}-1}}[1 - v - Kv + k_{\mathrm{r}}(K - Kv - v)]$, $D = -\frac{(K+1)(1-2v)}{2G}\left(\frac{\sigma_c}{k_{\mathrm{r}}-1} + P_0\right)$ and $K = \frac{1+\sin\psi_{\mathrm{r}}}{1-\sin\psi_{\mathrm{r}}}$.

(2) Solutions in softening region

The differential equations of the radial stress and of the plastic shear strain with respect to the original radius are derived as Equations (A4) and (A5).

$$\frac{\mathrm{d}\sigma_{\mathrm{r}}}{\mathrm{d}r} = \frac{1}{r}\left[(k_{\mathrm{p}}(\gamma^{\mathrm{p}}) - 1)\sigma_{\mathrm{r}} + \sigma_{\mathrm{c}}(\gamma^{\mathrm{p}})\right] \tag{A4}$$

$$\frac{\mathrm{d}\gamma^{\mathrm{p}}}{\mathrm{d}r} = \frac{-\frac{1}{2G}\left[(1-v)k_{\mathrm{p}}(\gamma^{\mathrm{p}}) - v\right]\frac{\mathrm{d}\sigma_{\mathrm{r}}}{\mathrm{d}r} - \frac{1}{r}\frac{1}{2G}\left\{\left[k_{\mathrm{p}}(\gamma^{\mathrm{p}}) - 1\right]\sigma_{\mathrm{r}} + \sigma_{\mathrm{c}}(\gamma^{\mathrm{p}})\right\} - \frac{\gamma^{\mathrm{p}}}{r}}{\frac{1}{2G}\left[(1-v)\sigma_{\mathrm{r}}\frac{\partial k_{\mathrm{p}}(\gamma^{\mathrm{p}})}{\partial\gamma^{\mathrm{p}}} + (1-v)\frac{\partial\sigma_{\mathrm{c}}(\gamma^{\mathrm{p}})}{\partial\gamma^{\mathrm{p}}}\right] + \frac{\partial\left[\frac{\gamma^{\mathrm{p}}}{1+K(\gamma^{\mathrm{p}})}\right]}{\partial\gamma^{\mathrm{p}}}} \tag{A5}$$

The displacement of rock mass material points in the softening region can be calculated by Equation (A6).

$$u = \frac{r}{2G}[(1-v)(\sigma_{\theta} - P_0) - v(\sigma_{\mathrm{r}} - P_0)] + r\frac{\gamma^{\mathrm{p}}}{1 + K(\gamma^{\mathrm{p}})} \tag{A6}$$

(3) Solutions in elastic region

The solution of the radial stress and the tangential stress in the elastic region can be expressed as Equations (A7) and (A8).

$$\sigma_{\mathrm{r}} = \frac{\sigma_{\theta}^{R_{\mathrm{p}}} + \sigma_{\mathrm{r}}^{R_{\mathrm{p}}}}{2} - \frac{\sigma_{\theta}^{R_{\mathrm{p}}} - \sigma_{\mathrm{r}}^{R_{\mathrm{p}}}}{2}\frac{R_{\mathrm{p}}{}^2}{r^2} \tag{A7}$$

$$\sigma_{\theta} = \frac{\sigma_{\theta}^{R_{\mathrm{p}}} + \sigma_{\mathrm{r}}^{R_{\mathrm{p}}}}{2} + \frac{\sigma_{\theta}^{R_{\mathrm{p}}} - \sigma_{\mathrm{r}}^{R_{\mathrm{p}}}}{2}\frac{R_{\mathrm{p}}{}^2}{r^2} \tag{A8}$$

where $\sigma_{\theta}^{R_{\mathrm{p}}}$ and $\sigma_{\mathrm{r}}^{R_{\mathrm{p}}}$ are the radial stress and tangential stress on the boundary between the elastic region and the plastic region ($R_p$).

The displacement of rock mass in the elastic region can be calculated by Equation (A9).

$$u = \frac{r}{2G}[(1-v)(\sigma_{\theta} - P_0) - v(\sigma_{\mathrm{r}} - P_0)] \tag{A9}$$

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
