# Peer review of "A New Numerical Finite Strain Procedure for a Circular Tunnel Excavated in Strain-Softening Rock Masses and Its Engineering Application"

_applsci, doi:10.3390/app12052706_

Round 1

Reviewer 1 Report

In this work the authors use finite strain theory in order to
calculate the mechanical response of a circular tunnel excavated in strain-softening rock masses. The developed procedure, based on the iteration and bisection method, sounds well developed from the theoretical point of view. Very interesting and useful the sensitivity analysis and the reliability design of Zhongyi tunnel. I suggest adding just one sentence to explain better the comparison between the results of the paper and Zhang’s in Fig. 7.

I do not have any concerns about the current version of this paper, for this reason, I believe it could be considered for publication.

Author Response

We are grateful for your pertinent and valuable comments and suggestions. As the suggestion of Reviewer 1, the below sentence is added in the second paragraph of section 5.1 to explain the comparison between the results of the paper and Zhang’s in Fig. 7.

“As shown in Figure 7, the GRC and the evolutions of softening region radius and residual region radius calculated by the proposed procedure are virtually identical with those of Zhang’s solution, which verifies the accuracy of the proposed numerical finite strain procedure.

Reviewer 2 Report

Dear authors,

The topic of the paper is interesting. Manuscript is generally well-written, logically organized, and adequately illustrated. The abstract is succinct and comprehensible. Introduction is relevant and theory based. The methods of study are generally appropriate, although clarification of a few details should be provided. In general, a good study was performed with obtaining promising results.

In the chapter 5. Verification, comparison, and discussion, (line 270) you have stated that the parameters of the MC rock mass listed in Table 1. are same with those in ‘Figure 6’ provided by study A numerical large strain solution for circular tunnels excavated in strain softening rock masses by Zhang et al. The problem is that the data in Table 1. used in this study and the data in Figure 6. from study provided by the Zhang et al. are not the same. Please check and correct this.

I also advise you to point out the advantages of the proposed new numerical finite strain procedure for circular tunnel excavated in strain-softening rock mass tunnel in relation to the existing ones. What is wrong with the current procedures and what are the benefits of applying the proposed procedures? This should be elaborated in the discussion and / or conclusion section.

Best regards and lots of success in future work.

Author Response

 The first problem of Reviewer 2 is explained clearly in ‘author-coverletter-17888940.v2.docx’.

As Reviewer 2 advised, the weakness of the current procedures is summarised more clearly in the fourth paragraph of the introduction section. The revised sentence is as follows. “The recursion method in [36-39] inevitably leads to the accumulative error which cannot be evaluated quantitatively and adjusted automatically. Increasing the number of concentric annuli can reduce the accumulative error only by an unknown extent and results in a loss of computation efficiency.”

And the benefits of the proposed new procedure are elaborated further in the end of section 5.1 and at the second paragraph of the conclusion section. In section 5.1, the performance of the proposed procedure is demonstrated as follows by the calculation example. “When the tolerances of the calculation relative error  are , , ,  and , the time for one whole calculation is 0.89s, 1.18s, 1.72s, 1.98s and 2.82s in this numerical example. It shows that for the proposed procedure the significant improvement of the precision does not affect the calculation efficiency excessively.”

In the conclusion section, the comparison between the two procedures is restated as follows. “The proposed procedure directly solves three sets of differential equations and uses bisection method to approximate the boundary conditions, realizing the quantitative control for the calculation error. More meaningfully, the highly efficient computing performance of the proposed procedure create conditions for its engineering application in the global sensitivity analysis and the reliability design of Zhongyi tunnel, which need large sample data.”
